# The Effects of Private Health Insurance on Universal Health Coverage Objectives in China: A Systematic Literature Review

**DOI:** 10.3390/ijerph17062049

**Published:** 2020-03-19

**Authors:** Runguo Wu, Niying Li, Angelo Ercia

**Affiliations:** 1Institute of Population Health Sciences, Barts and The London School of Medicine and Dentistry, Queen Mary University of London, London E1 2AB, UK; 2Global Health Policy Unit, School of Social and Political Science, University of Edinburgh, Edinburgh EH8 9LD, UK; 3School of Pharmacy, University of Wisconsin-Madison, Madison, WI 53705, USA; nli59@wisc.edu; 4Division of Informatics, Imaging & Data Sciences, School of Health Sciences, the University of Manchester, M13 9PL Manchester, UK; aenercia06@gmail.com

**Keywords:** private health insurance, healthcare financing, universal health coverage, China, access to healthcare, financial protection

## Abstract

Background: We conducted a systematic review on the role of private health insurance to complement the social health insurance system towards achieving universal health coverage in China. This review presents the impacts of private health insurance on expanding coverage, increasing access to healthcare, and financial protection. Methods: A systematic review was conducted by searching peer-reviewed articles published between January 2000 and March 2018 in Web of Science, PubMed, and China Knowledge Resource Integrated Database. The search terms included coverage prevalence, access and financial protection related to private health insurance in China. A total of 31 studies were selected. Results: Coverage prevalence of private health insurance gradually increased but it was unequally distributed across regions and populations. The expansion of social health insurance has enhanced the total aggregate premium of private health insurance but has had a mixed impact on the take-up of private health insurance. Private insurance beneficiaries were found to limit their utilisation of healthcare services and there was no evidence that it ensured financial protection. Conclusion: The role of private health insurance (PHI) in extending universal health coverage in China was limited and therefore should not be overstated.

## 1. Introduction

In the early 2000s, the Chinese government began to increase spending to expand the social health insurance system (SHI) after two decades of retrenchment. The public spending as the share of total healthcare expenditure in 2003 was 36.2% and increased to 56% by 2012 [1], which mostly came in the form of subsidisation for SHI contributions [2,3]. The increase in subsidies enabled people to gain coverage primarily from SHI schemes. Analysis of national data suggests that the overall health insurance coverage increased from 29.7% to 95.7% between 2003 and 2011 [4]. The three main SHI schemes in China–the Urban Employees’ Basic Medical Insurance (UEBMI), the Urban Residents’ Basic Medical Insurance (URBMI) for UEBMI-uncovered urban registrants, and the New Cooperative Medical Scheme (NCMS) for rural registrants–covered around 87% of the population [5]. Furthermore, government subsidies enabled coverage benefits to expand services in inpatient care, outpatient care, treatment of acute illness, and treatment of chronic disease [6].

Despite the development, many households are exposed to healthcare-related financial risks as SHI policies have limited coverage for treatment and prescriptions, co-payments, deductibles, and limited reimbursement [6] (Table 1). The annual incidence of catastrophic health expenditure continued to increase from 2003 at 12 to 12.9% in 2011 [4]. Furthermore, almost one-third of total health expenditure still came from out-of-pocket payments in 2013 [7].

Private health insurance (PHI) was introduced in China during the economic reforms of the 1980s. The Chinese government has increasingly considered the role of PHI as a financial source to its health system and address the coverage gap within the current SHI system [4,6]. Theoretically, the expansion of PHI could improve the current SHI by 1) being a substitute form of health coverage for individuals that are unable to get coverage from an SHI scheme; and 2) being a complementary and supplementary coverage for individuals covered by a plan in the SHI scheme. PHI prevalence was initially low and limited to corporate customers, who bought PHI for their employees. Since the mid-1990s, demand for PHI increased and more commercial insurers entered [12]. In 2003, there were only around three hundred PHI products in the market and no specialised PHI companies [12], while in 2013, more than one hundred commercial insurers and five specialised PHI companies operated PHI business, providing thousands of products [13].

In recent years, the PHI market has rapidly grown in terms of total aggregate premium income, whereas its compensation long accounted for a minor part of total health expenditure (Figure 1). Government officials especially aim to support SHI and PHI to become mutually reinforcing, such that PHI helps SHI to fill the gaps in coverage depth and height, and in turn the SHI expansion boosts the PHI market by increasing the awareness of the importance of health insurance and leaving sufficient room for PHI operation [13,14,15]. For example, the government has encouraged employers to purchase PHI for their employers in addition to being under the UEBMI scheme of SHI and implemented regulations to simplify the compensation process between PHI insurers and local SHI agencies [16]. Most recently, the government has offered tax incentives for employers and individuals purchasing PHI through a series of pilot programmes in several major urban centres [17].

While the Chinese government has increasingly considered extending the role of PHI, there are ongoing debates among global health scholars and policymakers over its effectiveness in expanding coverage and supporting the principles of universal health coverage (UHC). There is uncertainty on how effectively PHI can address the three dimensions of UHC with regards to the breadth (covered population), depth (covered benefits) and height (covered costs) of coverage [18]. An argument about the weakness of PHI’s ability to expand coverage is its principle that aims to meet demand—a combination of willingness and capacity to pay, which may put low-income populations at risk of not being able to obtain PHI coverage due to their inability to pay [19]. Certain groups might also be given risk-rated premiums, causing it to be unaffordable for this population to maintain and sustain their coverage [20].

Furthermore, the PHI market is vulnerable to market failures such as moral hazard and adverse selection. Moral hazard may contribute to patients using sub-optimal high levels of care, and physicians to prescribe unnecessarily expensive medications and more intensive treatments [21,22]. Insurers respond to moral hazard through co-payments and/or deductibles, therefore putting beneficiaries at risk for limited financial protection [23]. Adverse selection occurs as insurers have inaccurate information about the health status of enrolees. Insurers may set premiums based on average risks across the population (the community rating) [23]. The premium will be worth paying only for those with above average health risks, and therefore insurers may set the higher premium than the community rating, resulting in greater adverse selection. These factors may limit PHI’s ability to meeting principles of universal coverage [21,24], and could even exacerbate inequities in the society [25].

Advocates of PHI contend that PHI can potentially meet the goal of a social health insurance system to address current existing coverage gaps [26,27]. By introducing a different source of funding into the health system, the government can then better allocate resources to the population, improve weak administrative capability in the public sector [28], and solve limited public fiscal space for SHI [29]. They also argue that PHI is affordable to those who already rely on out-of-pocket payments for healthcare [24], and government subsidies can help those who cannot afford PHI [26]. The World Health Organization (WHO) and the World Bank have ambiguous views on extending coverage through PHI. They believe that PHI can contribute to expanding coverage prevalence and laying foundation for national insurance, but they also note its limitations [30].

In China, there have been some studies that associated PHI’s ability to address one or two issues with coverage gaps [14,31,32]. However, there is inconclusive evidence that suggest PHI can effectively extend overage and meet UHC principles. The purpose of our study was to conduct a systematic review that synthesises empirical evidence about PHI’s ability to play a role in financing China’s healthcare system and meet UHC principles. The review specifically focuses on understanding the role of PHI to expand coverage (in presence of expanding SHI), improve access to care, and provide financial protection. The current studies about PHI are still insufficient and fragmented. No study specifically and empirically associated the impacts of PHI with the three principles of UHC, and to our knowledge, no peer-reviewed systematic review on this topic has yet been published. To fill this gap, the review attempts to provide a broader understanding of how PHI have contributed to or limitedly contributed to these coverage targets by synthesising existing empirical evidence about PHI in China.

## 2. Materials and Methods

### 2.1. Search Strategy

This review focused on understanding the impacts of PHI on the UHC principles: Coverage prevalence, access to care, and financial protection [18]. The PRISMA guideline for systematic reviews [33] was adopted to design the search for peer-reviewed research articles in both English and Chinese that were published between January 2000 and March 2018. Three databases were selected: Web of Science (all databases), PubMed, and the China Knowledge Resource Integrated Database (CNKI), which is the largest, and most frequently updated database of Chinese-language academic publications.

We used the following search terms for all three synonyms of private health insurance in English: Private medical insurance, commercial health insurance, and commercial medical insurance. The terms used to refer to PHI in China were *shangye Jiankang baoxian,* and *shangye yiliao baoxian*. For the two English-language databases, the keyword “China” was included with each of the three search terms. This was not necessary for searches conducted in the CNKI database as all identified studies from CNKI are China-related.

Literature search was divided into three parts: 1) The coverage prevalence of PHI in China (including the impact of SHI on PHI coverage prevalence), 2) the effect of PHI on access to healthcare in China, and 3) the financial protection afforded by PHI in China. For coverage prevalence, the search terms were “prevalence”, “demand”, and “coverage” in English, and “xuqiu” (demand) or “fugai” (coverage) in Chinese. For access, the search terms were “access” or “utilisation/utilization” in English, and “keji” (access), “fuwu shiyong/liyong)” (service utilisation), “fuwu xuqiu” (service demand), “jiuyi” (using medical care), or “zhiliao” (treatment) were used in CNKI. For financial protection, the search terms “expenditure”, “expense”, “spending”, “payment”, or “cost” were used in the English language databases, while “zhichu” (expenditure), “huafei” (a less formal synonym of expenditure), “feiyong” (cost), and “jingjifudan” (financial burden) were used in CNKI. In addition, the citations of the included papers were scanned. If an article with a relevant title was cited but not identified in this search, it was added to the final review list after checking its eligibility using the criteria below.

The search strings used in Web of Science are presented here as an example of the searches used across the listed databases:

Search One: TS = ((health OR medical) AND (private OR commercial) AND (insurance china) AND (prevalence OR demand OR coverage));

Search Two: TS = ((health OR medical) AND (private OR commercial) AND (insurance china) AND (utilization OR utilisation OR access));

Search Three: TS = ((health OR medical) AND (private OR commercial) AND (insurance china) AND (expenditure OR expense OR spending OR payment OR cost)).

### 2.2. Inclusion and Exclusion Criteria

The following are the inclusion criteria for the studies selected for this review: (1) Empirical studies that were related to at least one of the three aspects of PHI, (2) that were conducted in China/Chinese health system, (3) that provided clear and full information of research design and methods, (4) that occurred between January 2000–March 2018, (5) and that were written in English and Chinese. The following are the exclusion criteria for the studies selected for this review: (1) Studies that examined willingness to buy PHI rather than enrolment or purchase, (2) that were based on tertiary data (i.e., those derived from other studies’ outcomes of analysing primary or secondary data) were not included, in order to avoid citing the same study repeatedly, (3) that did not examine one of the three aspects, i.e., coverage prevalence, access to care, and financial protection, (4) or that are editorials, commentaries, conference abstracts, book chapters, and discursive essays. Full text of the remaining papers was screened to exclude those that do not meet the inclusion criteria.

### 2.3. Quality Assessment and Risk of Bias

Two reviewers (R.W. and N.L.) independently assessed the quality of the studies included in the review using a quality-graded protocol with a 10-point scale system (see Appendix B). This appraisal tool was adapted from existing tools in previous relevant studies [34,35,36,37], and handbooks of systematic review [38,39] to suit this study. Appraisal results from two reviewers were compared. If the difference in the appraisal has no more than 1 point, then the average point was taken. The two reviewers discussed studies that had more than 1-point difference to reach consensus. Studies that were appraised as being low quality (0-3.5 points) were considered to have high risk of bias and therefore excluded from the review. Only studies that were appraised as being medium (4–6 points) and high (6.5–10 points) quality were reviewed in the study. The supplemental document (Appendix A) provides a more detailed account of the appraisal process. Risk of bias across studies was considered. Since for many of the reviewed studies, PHI is not their only research objective, and thus the result about PHI may not be crucial to publication, the publication bias on PHI should be moderate. To minimise the bias, we not only reviewed the text of included articles but also directly scanned all result tables and appendices to make sure all data about PHI, regardless of statistical significance, were extracted. This could avoid the omission of data about PHI that either do not have statistical significance or do not interest the authors.

## 3. Results

### 3.1. Characteristics of Included Studies

The initial search identified 692 studies (Figure 2). Forty-four studies met the inclusion criteria. Of the 44 included studies, the average score of the quality assessment was 5.10 out of 10 points. From the quality assessment, 16 were categorised as high-quality studies, 15 as medium-quality studies, and 13 as low-quality studies. All the low-quality studies were removed and a total of thirty-one studies were included in the review. Among the 31 studies, 24 studies examined the prevalence and distribution of PHI coverage (including 14 examined the relationship between PHI and SHI), 10 examined the effects of PHI on access to care, and nine examined the financial impacts of PHI (some studies are counted more than once due to engaging in more than one aspects of this review). For details of all these reviewed studies and the quality assessment results, see Table A1 (Appendix C).

Furthermore, 20 studies presented individual-level evidence, which focused on individuals’ behaviour of obtaining PHI, and individual use of healthcare or spending on healthcare. Nine studies used the China Health and Nutrition Survey (CHNS), an ongoing longitudinal household national survey that commenced in 1989. Five studies used the China Health and Retirement Longitudinal Study (CHARLS), a longitudinal national study that started in 2008 and became nationwide in 2011 and included people aged 45 and above. Two studies used the State Council’s URBMI Household Survey, ranging from 2007 to 2010; one used the Chinese Longitudinal Healthy Longevity Survey (CLHLS) that began in 1998 and studied people aged 65 years and above. The other studies used data from ad hoc, one-off surveys, most of which were primary data. Additionally, 11 studies focused on aggregate level evidence. The most common data source used were provincial-level statistics provided by government departments. Another source of aggregate data was from the National Health Service Survey (NHSS), which provides the county-level data.

Nine individual-level studies examined longitudinal data, and the other 11 individual-level studies examined cross-sectional data. In terms of aggregate-level studies, all but one examined longitudinal data. All survey databases used in these studies, including CHNS, CHARLS, NHSS, the URBMI survey, CLHLS and government provincial-level statistics, included multiple Chinese provinces. For one-off surveys, the scope of data collection varied depending on the objectives of the studies.

Twenty-nine out of the 31 studies employed regression models as their principal methods of data analysis, adjusting for individual and aggregate background factors. Generally, these regression models included basic linear, logit, or probit models for cross-sectional data, and fixed effects, random effects, or dynamic models for longitudinal data. Some studies introduced instrumental variables to treat endogeneity. Other studies included difference-in-difference estimators to extract policy effects by comparing treatment and control groups. A few studies used the Heckman models or Two-part models to handle sample selection.

### 3.2. Coverage Prevalence

#### 3.2.1. Trend and Distribution

As there is no official data about the population coverage rates of PHI from the China Insurance Regulatory Commission or other state regulatory or statistical agencies, all such estimates were derived from surveys. Studies examining nationwide representative samples (those include multiple provinces and both urban and rural areas) reported the lowest coverage rate of 4% between 2004 and 2006 and the highest coverage rate at 12% recorded in 2013 [32,40,41,42]. This suggests that PHI coverage prevalence gradually increased over time. Those that focused on the latest urban samples reported the highest coverage rates: 19% [43] and 35% [44], respectively, from two studies using relatively small urban samples. Instead of population coverage, the China Insurance Regulatory Commission publishes Insurers’ aggregate premium income of selling PHI policies (equal to customers’ spending on buying PHI aside of taxes) at the provincial level every year, and all related studies in this review used the statistics. The total PHI premium income substantially increased by 1629% around the same time period, from ¥6.5 (≈ US$0.9) billion in 2000 to ¥112.4 (≈ US$15.7) billion in 2013 [45].

In terms of the distribution (Table 2), one study examined the 2006 CHNS data and found that rural residents in eastern provinces were more likely to have PHI than their inland counterparts, other things being equal [46]. Another study found that eastern provinces were associated with greater income from PHI premiums per capita than central or western inland provinces, other things being equal [47]. Two nationwide studies found that urban residents were more likely to be covered by PHI than rural residents, other things being equal [41,48]. Another multi-province study found that students living in urban areas were more likely to have PHI than non-students, but this phenomenon was found to disappear in rural areas [32]. One study that focused on three eastern affluent cities of Beijing, Shanghai, and Xiamen, found no significant difference in PHI coverage between urban and rural residents [44]. Additionally, one study showed that individuals who migrated from rural areas to urban areas were either more likely to be insured by PHI or completely uninsured than urban locals, other things being equal [41].

#### 3.2.2. Relationship between PHI and SHI Coverage

There is mixed evidence with regards to the impact of SHI enrolment on obtaining PHI. Two studies supported a positive correlation between PHI enrolment and SHI enrolment [32,46]. However, two studies found a negative correlation [41,42], and one study reported a neutral (insignificant) relationship [40] (Table 3). Additionally, one study found that the NCMS (rural SHI scheme) membership increased adults’ PHI enrolment, but decreased enrolment among children, particularly among lower income groups [14]. Another study found that enrolment in the SHI schemes (UEBMI, URBMI, and NCMS) was negatively associated with enrolment in PHI in the whole population. However, disaggregating the data suggests that the negative correlation was only significant in urban areas [41]. Another study found that enrolment in the NCMS had a negative correlation with PHI enrolment between 2004 and 2006, but the correlation became positive between 2006 and 2009 [49].

On the contrary, all studies that examined the provincial-level correlation of insurers’ income from PHI premium with SHI scheme coverage indicators concluded that SHI expansion increased insurers’ income from PHI premium, other things being equal [47,50,51,52,53,54,55]. These studies examined a range of different regions, time periods, and SHI schemes. Five made regression models that controlled for demographic factors (e.g., average age, gender ratios, affluence, and education levels) [47,50,51,52,53], and two used other methods [54,55].

### 3.3. Access to Healthcare Services

The findings of this review on access to healthcare were mainly informed by utilisation of healthcare services. For generic utilisation (including utilisation of both outpatient and inpatient care), the evidence is mixed. One study found that having PHI had no impact on utilisation compared with people without PHI coverage [56]. Another study found that having PHI increased utilisation compared with those without PHI for urban population but not for rural population, other things being equal [57]. The third study found that having PHI increased utilisation compared with those covered under the NCMS, other things being equal [58] (Table 4).

There is some evidence to suggest that having PHI enabled beneficiaries to utilise inpatient services, other things being equal (Table 4). Specifically, one study found that enrolment in PHI was positively associated with the utilisation of inpatient care [59]. Another study found that enrolment in PHI was associated with increasing lengths of hospitalisation between 2000 and 2004, but the association was not significant between 2006 and 2009 [60]. However, a study that focused on rural-to-urban migrants found no significant association of PHI enrolment with the migrants’ inpatient utilisation [61]. At the aggregate level the findings show no significant relationship between PHI coverage prevalence and the average utilisation of inpatient services [62,63], suggesting that PHI minimally benefit inpatient utilisation for the population as a whole.

By contrast, there is minimal evidence that suggests PHI enrolment increased the utilisation of outpatient services, other things being equal (Table 4). Only one study reported a positive relationship between enrolment in PHI and outpatient utilisation [64]. However, in this study, the PHI was restricted to substitutive health insurance (enrolled in PHI only). There was no significant correlation on the effect of having complementary PHI (the marginal effect of PHI enrolment for those enrolled in both PHI and SHI) on utilising outpatient services. In addition, the single aggregate-level study reported that the percentage of PHI enrolees had a positive relationship with the average number of outpatient visits of rural counties [63].

Although few PHI policies in China cover preventative care (i.e., physical examinations and vaccinations) [65], three studies found that PHI enrolment had a significant positive relationship with the utilisation of preventative services [60,61,66].

### 3.4. Financial Protection

The findings of this review suggest that having PHI did not reduce out of pocket payments. Two studies found that enrolment in a PHI plan was not associated with reduced out-of-pocket payments (Table 5) [56,67]. One study found that PHI increased the out-of-pocket share of total health expenditure among high-income populations but had no such effect among low-income populations between 2000 and 2004 [60]. The other study found that enrolment into PHI was associated with higher chances of out-of-pocket payments of more than ¥1000 (≈ US$140), and more than ¥5000 (≈ US$700), compared to out-of-pocket payments of less than ¥1000 [44]. Only one study found that PHI reduced out-of-pocket payments for those covered solely by PHI but did not reduce out-of-pocket payment for those covered by both PHI and SHI [67].

While out-of-pocket healthcare expenditure indicates financial risk, total healthcare expenditure, measured by gross healthcare payments before insurance reimbursement, reflects the financial burden of the health system more than it indicates individuals’ financial risk [44,68]. Four studies found that enrolment in PHI increased total healthcare expenditure of the enrolees (Table 5) [42,44,57,69]. No individual-level opposite evidence exists. Only two studies looked at the financial impacts of PHI at the aggregate data. One found that the percentage of PHI coverage had no significant impact on the per-capita annual medical expenditure at the county level [63]. The other found that per-capita PHI premium spending significantly associated with less per-capita medical expenditure at the provincial level [70].

## 4. Discussion

In the past 20 years, the publicly managed SHI helped China move towards expanding health insurance coverage to its population. However, the extent to which the depth (services) and height (costs) of coverage was still limited due to restricted government’s financial capacity [4,6], which is also an international problem for many countries on the way to approach UHC [28,29]. For policy makers, the main goal of introducing PHI was to further assist with the expansion of SHI and extend the depth and height of coverage [13,27].

This systematic review found that the coverage prevalence of PHI gradually increased since 2000, while commercial insurers’ income from PHI increased to a greater extent [45]. This suggests a rapid increase in the cost of PHI and an upmarket movement of China’s PHI market. Additionally, in China’s PHI market, many PHI plans are sold as part of a bundle package including other savings products or life and accident insurance products [71], so it is possible that insurers’ income from the whole product bundle is counted as income from PHI premium, pushing up its costs. As a result, the increase of insurers’ income from PHI premium did not go along with a substantially larger number of people covered by PHI.

It is still unclear whether the expanding SHI coverage boosted or suppressed the coverage prevalence of PHI. The direct evidence about the impact of SHI enrolment on the uptake of PHI is mixed. On the contrary, there is strong evidence that the SHI expansion was associated with insurer’s income increase from PHI premium, controlling for economic growth and relevant population characteristics, but the evidence is indirect since insurer’s income increase from PHI premium does not equal the increase in population coverage of PHI as stated above.

One theory suggests that public health insurance programmes can crowd out PHI due to the duplication of benefits [72,73]. By contrast, several scholars have argued that public health insurance expansion may help boost the coverage prevalence of PHI. According to [15], the limited coverage of SHI in China can cause PHI insurers to lower their cost and introduce additional plans to attract the uninsured to purchase coverage. Meanwhile, the introduction of SHI helps disseminate knowledge about health insurance in countries where awareness of insurance is rare, increasing the demand for PHI [13,14]. As a result, SHI could theoretically cause the premium income of PHI to increase. It is, however, unclear the extent of this effect on the coverage prevalence of PHI in the Chinese context, and this needs further investigation. The review found two studies suggesting a positive correlation between having PHI and the utilisation of inpatient care while one study on rural-to-urban migrants found the correlation being neutral (Table 4). However, little evidence suggests having PHI affect the use of outpatient care. This is supported by findings from studies in other countries [74,75,76]. In China, the benefits package of PHI given to beneficiaries usually includes protection for critical diseases, compensation for hospitalisation, and access to superior amenities, such as VIP (premium wards, better services, etc.) [65,71]. However, due to price control and actuarial difficulties, most PHI plans limitedly provide coverage for outpatient services and medications [77,78]. This study also found a positive correlation between PHI enrolment and the utilisation of preventative services, even if few PHI plans tend to include them [65]. A possible explanation for this is that commercial insurers offer additional benefits for individuals that take advantage of using preventative services.

There is no evidence that suggests PHI can reduce out of pocket expenses for beneficiaries. PHI plans include various levels of deductibles and benefit packages. For example, many PHI policies compensate costs of hospitalisation but restrict covering medication costs. However, several studies have found that the majority of out of pocket expenses is attributable to medication costs [77,79,80]. Therefore, financial protection of PHI was compromised. There have been findings from Brazil and South Korea that also suggest that enrolment into PHI did not reduce out of pocket expenses on healthcare [81,82].

There is evidence that PHI increased individual total health expenditure. This could cause more financial burden to governments, as a part of the expenditure must be reimbursed by insurers and insurers often receive governments subsidies for PHI. For the reasons of raising total health expenditure, in addition to utilisation increased by PHI, commercial insurers may have less bargaining power over the price of care than public insurers, particularly in a single-payer system such as the SHI system in China [20], further pushing up total health expenditure. There has been supporting evidence from foreign countries [22,83].

In order to improve PHI’s depth of coverage and providing financial protection, the government may want to define the mandated benefits package. For example, the governments of the Netherlands, South Africa, and to an extent, the United States, have required PHI policies to provide basic services to all their beneficiaries [84,85,86]. However, on one hand, effectively determining what type of covered benefits included in PHI policies needs careful consideration, and to an extent exceed the capacity of regulations. On the other hand, such policy lends itself to subsidisation. Whether it is worth subsidising PHI rather than expanding investment in SHI is open to question [87].

There was moderate evidence that suggests the distribution of PHI in China was unequal and favoured the relatively affluent urban and eastern areas. Taking into account the well-documented pro-rich demand for PHI at the individual level together [21,22,23], our findings thus suggest that PHI is not an effective form of coverage for low-income populations, especially for those living in less affluent areas, where there might be fewer PHI selling agencies, lower availability of PHI information, and poorer connection between insurers and healthcare providers, since PHI sellers tend to cluster in densely populated affluent urban areas for a prudent strategy [75].

Consequently, a big challenge to implementing PHI as a means for improving UHC is expanding its coverage prevalence and meanwhile protecting equity. As profitability of the PHI market in China is still questionable [13], commercial insurers hesitate to expand business to attract less affluent population groups [65]. Government subsidisation or tax break may help its expansion. However, the benefits mainly flow to the more affluent groups, as they are more likely to have the financial means to purchase PHI. In addition, affluent regions in China may introduce additional subsidies to its residents compared to less affluent regions, thus increasing inequalities with regards to accessing PHI across the country. Scholars have suggested strong government regulations against voluntary enrolment and risk-pricing of PHI to promote equal access to PHI [23]. For example, in countries like Uruguay and Switzerland, the governments mandate the purchase of PHI [87], and in the Netherlands and Chile, pricing of some PHI policies is income-related rather than risk-related [84,87].

This study has two suggestions for future research based on the findings. First, given the inconsistency between aggregate PHI premium income and coverage prevalence, to examine the coverage contribution of PHI, only analysing premium income data is at the risk of being misleading. Nevertheless, most policy articles concerning the present development and prospect of PHI in China only used premium income data [12,13,88,89] possibly because it is relatively available and easy to use (there is insofar no official PHI take-up or population coverage data in China, except those derived from the surveys as previously mentioned).

Second, PHI plans, as voluntary for-profit health insurance schemes, are intrinsically different from SHI schemes, as SHI’s objectives are set at the system level and prioritize people’s health needs [25]. The nature of PHI, including enrolment that is based on capacity to pay, risk pricing, and limited population coverage, raises the concern that it could undermine the essential equity objective of UHC when it benefits the enrolees at the expense of others through relocating limited health resources according to membership rather than need [20,25]. However, this review found that the aggregate-level evidence that addresses health equity questions is limited and tends to be inconsistent with those from the individual level. More studies at the aggregate level is needed.

This review has several limitations. There were only a limited number of studies identified in this topic and the data used is fragmented. Therefore, it was not possible to extract the data and aggregate them to conduct a meta-analysis or any quantitative analysis. Instead, this study aimed to collate the studies on this topic and present them in a narrative way, in order to better understand how PHI has been working from the perspectives of addressing the UHC objectives in China. Second, most of the reviewed studies relied on regression models on survey data. There was a lack of experimental studies. Although many of these studies suggested causality in discussion, the results, which are basically associational, need to be interpreted with caution. Third, all the studies used quantitative data with different methodologies and data sources. This review differentiated them mainly according to validity and reliability of methods and data used, and subsequently only reviewed medium- and high-quality studies. It is impossible to rule out useful information in the low-quality group. Lastly, this review included limited studies that present the current situation in China. However, it is able to reflect on PHI’s contribution to the healthcare system in place, because the basic modality of China’s healthcare financing, as mentioned in the introduction, has not changed since the late 1990s [3,13], and hence, data in the 2000s-2010s remain relevant to the current situation.

## 5. Conclusions

To our knowledge, this is the first systematic literature review that focused on understanding the Chinese government’s decision to expand the role of PHI to pursue UHC objectives. This review found that PHI coverage prevalence has increased moderately in China. However, the growth of PHI premium income was higher compared to take-up of the coverage type. The distribution of PHI enrolees favoured people residing in relatively affluent eastern and urban areas. Whether the expanding SHI coverage boosted or suppressed the coverage prevalence of PHI is still unclear. The contribution of PHI to depth of coverage was limited. Evidence suggests that it had little impact on the use of outpatient care and a mixed impact on the use of inpatient care. Coverage of PHI did not demonstrate financial protection, because it increased total health expenditure but did not reduce out-of-pocket health expenditure. In sum, it suggests that PHI’s contribution to extending UHC in China has been limited and therefore should not be overstated. Government subsidisation, mandated benefits packages, and strong regulations on the PHI market may help PHI to play a more effective role in the progress towards UHC.

## Figures and Tables

**Figure 1 ijerph-17-02049-f001:**
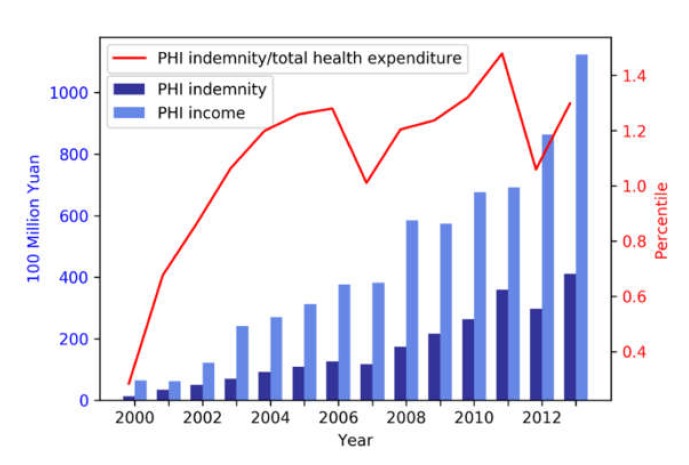
Income and indemnities of private health insurance (PHI), and the share of PHI indemnities in total health expenditure in China. Data source: Yearbook of China’s Insurance 2014 (1 yuan or ¥1 ≈ US$0.14 at the current exchange rate. The following conversations referred to this rate).

**Figure 2 ijerph-17-02049-f002:**
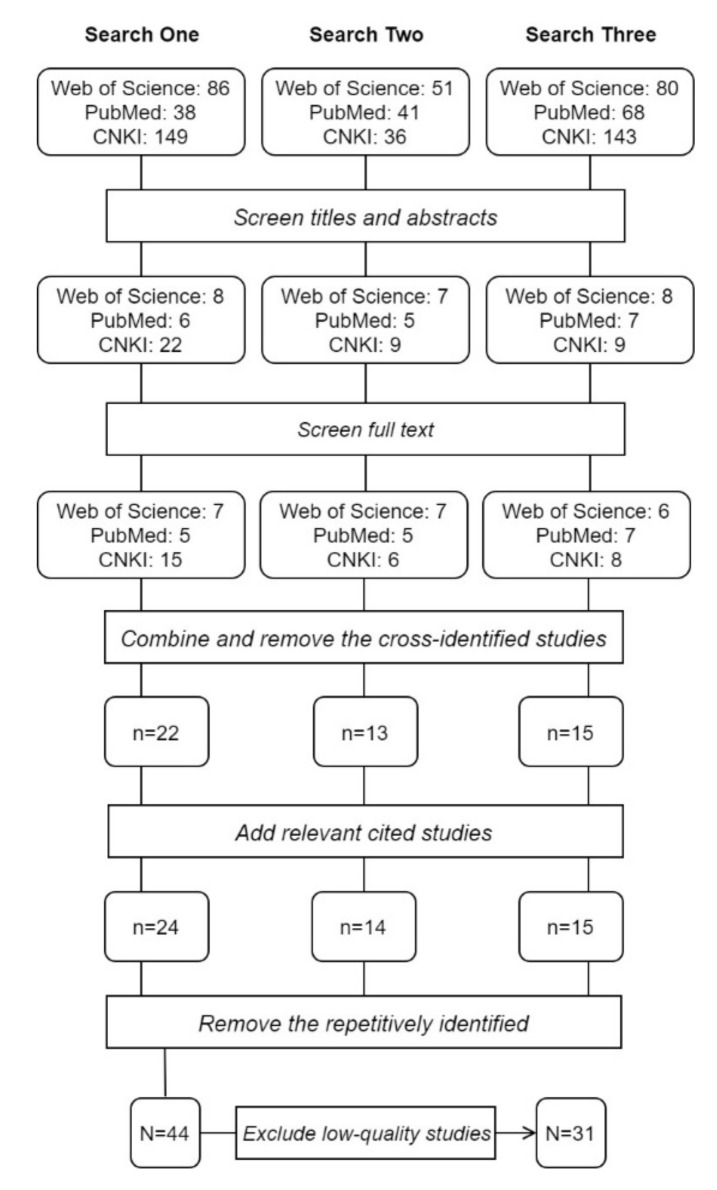
The flowchart of the systematic searches and selection.

**Table 1 ijerph-17-02049-t001:** Summary of China’s social health insurance institutions ^1^.

SHI Schemes ^2^	UEBMI	NCMS	URBMI
Year of launch	1998	2003	2007
Administration department	Human Resource and Social Security	Health	Human Resource and Social Security
Target population	Urban employees	Rural registrants	Urban registrants without UEBMI
Pooling level	Prefecture	County	Prefecture
Number of pools	333	2852	333
Enrolment	Compulsory	Voluntary	Voluntary
Number of members	265 million	805 million	272 million
Individual contribution	2–3% of salary	¥30–50	¥30–50
Employer/government contribution	6–8% of salary	¥200	¥200
Inpatient reimbursement rate	81%	< 50%	64%
Outpatient reimbursement rate	Depends on MSA	0–40% ^3^	0–40% ^3^
Reimbursement cap	Six-times local average wage	Eight-times local peasants’ income	Six-times local disposable income

^1^ Based on 2011–2012 data. ^2^ SHI = social health insurance; UEBMI = Urban Employees’ Basic Medical Insurance; NCMS = New Cooperative Medical Scheme; URBMI = Urban Residents’ Basic Medical Insurance; FMS = Free Medical Scheme; MSA = UEBMI’s individual medical savings account. ^3^ An approximate estimate as the coverage of NCMS and URBMI gradually expanded and varied spatially; the data about UEBMI, NCMS, and URBMI refer to [3,6,8,9,10,11].

**Table 2 ijerph-17-02049-t002:** Comparison of PHI coverage spatially or between different migration status ^1^.

Ref. ID	Study period	Comparison	Results
[47]	2007–2013	East vs. inland	East provinces were associated with higher PHI premium income
[46]	2006		For rural residents, living in the east was associated with a higher chance of enrolment into PHI
[48]	2011	Urban vs. rural	Living in urban areas was associated with a higher chance of enrolment into PHI
[41]	2011, 2013		Living in urban areas was associated with a higher chance of enrolment into PHI
[44]	2011		Living in urban areas was NOT significantly associated with a higher chance of enrolment into PHI
[32]	2000, 2004, 2006		For students, living in urban areas was associated with a higher chance of enrolment into PHI
[41]	2011, 2013	Migrants vs. locals	Rural-to-urban migrants was associated with a higher chance of enrolment into PHI

^1^ If unspecified, all comparisons passed the significance test.

**Table 3 ijerph-17-02049-t003:** Relationship between PHI prevalence and social health insurance(SHI) system expansion ^1^.

Ref. ID	Study Period	PHI Indicator	SHI Indicator	SHI Schemes ^2^	Sample	Correlation
	Aggregate Level Evidence					
[47]	2007–2013	Income	Percentage of enrolees	All	Mixed	Positive
[50]	2002–2007	Income	Percentage of enrolees	All	Mixed	Positive
[51]	2000–2007	Income	Fund income	UEBMI & URBMI	Mixed	Positive
[52]	2002–2009	Income	Fund income	UEBMI	Mixed	Positive
[53]	2003–2012	Income	Average compensation	UEBMI & URBMI	Mixed	Positive
[54]	2005–2010	Income	Fund income	All	Mixed	Positive
[55]	2005–2011	Compound index ^3^	Compound index ^4^	NCMS	Mixed	Positive
	Individual Level Evidence					
[32]	2000,2004,2006	Enrolment	Enrolment	Urban schemes	Urban	Positive
[46]	2006	Enrolment	Enrolment	NCMS	Rural	Positive
[42]	1989–2009	Enrolment	Enrolment	All	Mixed	Negative
[41]		Enrolment	Enrolment	All	Mixed	Negative
	2011, 2013				Urban	Negative
					Rural	Neutral
[40]	2004–2011	Enrolment	Enrolment	URBMI	Urban	Neutral
[14]	2011	Enrolment	Enrolment	NCMS	Adult	Positive
					Child	Negative
[49]	2004, 2006, 2009	Enrolment	Enrolment	NCMS	Rural	Negative
						Positive

^1^ If unspecified, all presented positive or negative correlations passed the significance test, otherwise neutral correlation was reported. ^2^ UEBMI = Urban Employees’ Basic Medical Insurance; URBMI = Urban Residents’ Basic Medical Insurance; NCMS = New Cooperative Medical Scheme. ^3^ Index generated by income, expenditure, claim ratio, etc. ^4^ Index generated by income, expenditure, ratio of income and expenditure, etc.

**Table 4 ijerph-17-02049-t004:** The correlation between PHI and access to healthcare ^1^.

Ref. ID	Study Period	PHI Indicator	Type of Healthcare Utilised	Sample	Correlation
[56]	2004	Enrolment	Generic healthcare	Mixed	Neutral
[58]	2008	Enrolment	Generic healthcare	Mixed	Positive ^2^
[57]	2008	Enrolment	Generic healthcare	Mixed/urban/rural	Positive/Neutral ^3^
[60]	2000, 2004	Enrolment	Inpatient care	Mixed	Positive ^4^
			Preventative care		Positive
[59]	2007, 2008	Enrolment	Inpatient care	Urban	Positive
			Outpatient care		Neutral
[61]		Enrolment	Inpatient care	Rural-to-urban migrants	Neutral
	2007–2010		Outpatient care		Neutral
			Preventative care		Positive
[66]	2004, 2006, 2009	Enrolment	Outpatient care	Rural	Negative
			Preventative care		Positive
[64]	2011, 2013	Enrolment	Outpatient care	Mixed	Positive/Neutral ^5^
[62]	2006–2010	Provincial PHI premium income over GDP	Inpatient care (the average length of hospitalisation)	Mixed	Neutral
[63]	2003	Percentage of PHI enrolees in a county	Inpatient care (the number of admissions per 1000 in 52 weeks)	Rural	Neutral
			Outpatient care (the number of visits per 1000 in 2 weeks)		Positive

^1^ If unspecified, all comparisons are between enrolees and non-enrolees of PHI, and all presented positive or negative correlations passed the significance test, otherwise neutral correlation was reported. ^2^ Referring to the NCMS (New Cooperative Medical Scheme). ^3^ For the whole and the urban population, not for the rural population. ^4^ The positive relationship exists between 2000 and 2004 but disappears between 2006 and 2009. ^5^ Positive for PHI as primary health insurance only; no correlation for complementary PHI.

**Table 5 ijerph-17-02049-t005:** The correlation between PHI and financial risk ^1^.

Ref. ID	Study Period	PHI Indicator	Financial Risk Indicator	Sample	Correlation
[56]	2004	Enrolment	Out-of-pocket payments	Mixed	Neutral
[67]	2011–2012	Enrolment	Out-of-pocket payments	Mixed	Neutral/Negative ^2^
[60]	2000, 2004	Enrolment	Out-of-pocket payments (as a share of total health expenditure)	Mixed	Positive/Neutral ^3^
[44]	2011	Enrolment	Out-of-pocket payments exceeding ¥1000 and ¥5000	Urban	Positive for both
			Total health expenditure exceeding ¥1000		Positive
[69]	2003, 2005	Enrolment	Total health expenditure	Urban	Positive ^4^
[57]	2008	Enrolment	Total health expenditure	Mixed/urban/rural	Positive/Neutral ^5^
[42]	1989–2009	Enrolment	Total health expenditure	Mixed	Positive
[63]	2003	Percentage of PHI enrolees in a county	Per-capita health expenditure	Rural	Neutral
[70]	2006–2012	Provincial per-capita PHI premium income	Per-capita health expenditure	Mixed	Negative

^1^ If unspecified, all comparisons are between enrolees and non-enrolees of PHI, and all presented positive or negative correlations passed the significance test, otherwise neutral correlation was reported. ^2^ Neutral for all PHI and complementary PHI, and negative for PHI as primary health insurance only. ^3^ Only positive for the high-income group between 2000 and 2004, but neutral for the low-income group and all groups between 2006 and 2009. ^4^ Comparing to SHI. ^5^ Positive for the whole and the rural population, but neutral for the urban population.

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
