# Peer review of "The Effects of Private Health Insurance on Universal Health Coverage Objectives in China: A Systematic Literature Review"

_ijerph, 2020, doi:10.3390/ijerph17062049_

Round 1

Reviewer 1 Report

The paper is well-organized and provides a nice overall assessment of PHI in China. The approach to selecting papers for review is explained clearly. I have only a few minor concerns:

"Access" is addressed only through studies of utilization. What has been the trend in the supply of medical services and/or providers, hospitals, certain specialties?

How many insurers provide PHI and has the market been stable? Profitable?

The authors could devote a little discussion to the theoretical expectations and findings from other countries. 

Author Response

Dear reviewer,

Thank you very much for your kind feedback and suggestions to improve our article. It is our pleasure to respond to your comments and reflect them in the article.

Please see the attachment to know how we have responded to your concerns. We hope the responses address your concerns properly.

Sincerely,

Runguo Wu

Reviewer 2 Report

I am very pleased to review interesting article. I think your systematic review provided empirical evidence about how China's PHI affected UHC. However, it is regrettable that the selected studies are not published currently, I am concerned that the results may not be consistent with the current situation.

A few points need to be clarified as detailed below.

Line 22-24, 67-69, 360-361, 440-441

- I think it is important to understand the relationship between SHI and PHI coverage. So it is necessary to more detailed explain the reason why SHI expansion boosts the PHI market.

Line 366-367, 442-443

- Can you conclude by this systematic review that PHI promotes the use of inpatient care in China? Please explain more detail about this point

Table 2~5

- It is recommended to indicate the study period of each Ref ID.

- There is no mention of <table2> in the article. Please indicate <table2> in the article.

Other

- In China, it was concluded that PHI contributed limitedly to the expansion of UHC and should not be overstated. What changes do you think PHI needs to make more contributions to China's UHC? I recommend containing this issue to discussion or conclusion.

Author Response

(The authors gave the same response as above.)
